# Association of a Global Invasive Pest *Spodoptera frugiperda* (Lepidoptera: Noctuidae) with Local Parasitoids: Prospects for a New Approach in Selecting Biological Control Agents

**DOI:** 10.3390/insects15030205

**Published:** 2024-03-19

**Authors:** Ihsan Nurkomar, Ichsan Luqmana Indra Putra, Damayanti Buchori, Fajar Setiawan

**Affiliations:** 1Department of Agrotechnology, Faculty of Agriculture, Universitas Muhammadiyah Yogyakarta, Yogyakarta 55183, Indonesia; ihsan.nurkomar@umy.ac.id; 2Department of Biology, Faculty of Applied Science and Technology, University of Ahmad Dahlan, Yogyakarta 55166, Indonesia; 3Department of Plant Protection, Faculty of Agriculture, IPB University, Bogor 16683, Indonesia; 4Center for Transdisciplinary and Sustainability Sciences, IPB University, Bogor 16153, Indonesia; 5Laboratory of Geographic Information System and Remote Sensing, Research Center for Limnology and Water Resources, National Research and Innovation Agency (BRIN), Kabupaten Bogor 16911, Indonesia; faja006@brin.go.id

**Keywords:** *Brachymeria*, Eupelmidae, host–parasite interaction, local adaptation, Platygasteridae, *Telenomus remus*, *Stenobracon*

## Abstract

**Simple Summary:**

The classical biological control approach is generally used in biological control practice. Through this research, we reveal the findings of a new association between *S. frugiperda*, an invasive corn pest in several nations, including Indonesia, and local parasitoids, suggesting the new association approaches in biological control practices. *Telenomus remus* is the most dominant egg parasitoid of *S. frugiperda* found in the field, suggesting their quick adaptation to new pests.

**Abstract:**

*Spodopotera frugiperda* is a worldwide invasive pest that has caused significant economic damage. According to the classical biological control approach, natural enemies that can control invasive pests come from the same area of origin as the pests that have experienced coadaptation processes. However, the new association’s approach suggests that local natural enemies are equally capable of controlling invasive pests. Due to the lack of data on the association of *S. frugiperda* and local natural enemies, research was conducted through a rapid survey to study the diversity of parasitoids associated with *S. frugiperda*. The results showed 15 parasitoid species associated with *S. frugiperda*. Four egg parasitoids, eight larval parasitoids, and three larval–pupal parasitoids were found to be associated with *S. frugiperda* for three years after it was first discovered in Indonesia. Eleven of them are new reports of parasitoids associated with *S. frugiperda* in Indonesia. A new association was found between *S. frugiperda* and twelve parasitoid species, consisting of three egg parasitoids (Platygasteridaesp.01, Platygasteridaesp.02, and *Telenomus remus*), six larval parasitoids (*Apanteles* sp., *Microplitis* sp., *Campoletis* sp., *Coccygidium* sp., *Eupelmus* sp., and *Stenobracon* sp.), and three larval–pupal parasitoids (*Brachymeria lasus*, *B. femorata*, and *Charops* sp.). *Telenomus* remus is the most dominant parasitoid, with a higher abundance and parasitism rate. The result suggests another method for selecting biological control using the new association approach since local natural enemies can foster quick adaptation to invasive pests.

## 1. Introduction

The appearance of invasive pests is a problem that requires attention because it may threaten agriculture and the variety of local species [1] and cause biotic homogenization [2]. *Spodoptera frugiperda* J.E. Smith (Lepidoptera: Noctuidae) is an invasive pest from America. It has become a new pest in Indonesia since early 2019 [3]. This pest has a wide range of distribution. It has now spread to 32 provinces in Indonesia, including Sumatra [4], Java [5], Kalimantan [6], and Sulawesi [7]. *Spodoptera frugiperda* infestations should be severely considered since a population of 0.2 to 0.8 *S. frugiperda* larvae/plant can decrease maize productivity by 20 to 50% [3]. Reports of damage due to *S. frugiperda* have been reported in several countries, such as Ethiopia and Kenya (32–47%) [8], Zimbabwe (32–48%) [9], Ghana (22–67%) [10], and Indonesia (60%) [11]. *Spodoptera frugiperda* has reportedly replaced the position of Asian corn borer *Ostrinia furnacalis* Guenée (Lepidoptera: Crambidae) as the primary pest of maize in China [12]. Rizali et al. [2] mentioned that the presence of *S. frugiperda* significantly decreases the intensity of attack of other lepidopteran pests and indirectly causes negative effects on the diversity of their natural enemies (particularly predators) in different maize fields in Indonesia.

A strategy to control *S. frugiperda* can be created using natural enemies such as parasitoids [13] to enhance the currently used control strategies, including the use of genetically modified varieties and the overuse of pesticides [14]. Generally, classical biological control (CBC) involves searching for natural enemies in the pest’s native area, known as the old association [15], a particularly effective method for managing invasive insects that spread widely and infiltrate various habitats. The CBC program for controlling *S. frugiperda* in its area of origin has been carried out, for example, the introduction of *Archyas incertus* from Argentina, *Eiphosoma vitticolle* from Bolivia, and *Cotesia marginiventris* from the US to the Caribbean [16,17]. However, no information has been published regarding the deployment of CBC agents against *S. frugiperda* within its invasion range [13].

On the other hand, Hokkanen and Pimentel [18] proposed the possibility of new associations between herbivores and local natural enemies. One of the arguments that has been proposed is because a new association that can be established usually inflicts extreme damage on the new host [19]. Further Pimentel [20] argues that “old association” is linked to ecological homeostasis, thus giving some reasons why parasites from native habitats sometimes do not provide the expected control. In the first half of the 20th century, at least ten parasitoids from different species of *Spodoptera* were collected on other continents and introduced in different American countries [16,17]. But only one species—the egg parasitoid *Telenomus remus*—was introduced to the Americas from India, established itself, and expanded throughout *S. frugiperda’s* entire distribution range in the Americas. Although *T. remus* has never proved an effective natural enemy in the Americas, it has been widely employed as an augmentative biological control agent [21,22]. In contrast, *S. frugiperda* has significant parasitism rates in areas it has invaded, such as Africa and Asia, with more than 60 associated local species of parasitoids [23]. Compared to the Americas, these regions had greater parasitism rates of eggs by *T. remus*—over 50% in some East Africa [24], 26% in Ghana, 14% in Benin [25], and 30% in China [26,27]. Navik et al. [28] stated that *Trichogramma chilonis* parasitizes 16–24% of *S. frugiperda* eggs in India. It is also rather common in China [27] and Africa [24]. Agboyi, Goergen, Beseh, Mensah, Clottey, Glikpo, Buddie, Cafà, Offord, and Day [25] also reported that parasitism of *S. frugiperda* larvae in Ghana ranged from 5% to 38% and 13% to 53% in East Africa [24]. These findings suggest interesting research questions, such as how *S. frugiperda* and local parasitoids interact in a high-biodiversity country like Indonesia.

Several studies have been reported since *S. frugiperda* was first reported in Indonesia. These studies were primarily focused on the presence/absence, diversity, infestation level, and ecology of *S. frugiperda* [5,29,30,31,32]. Research on the performance of local parasitoid species in Indonesia in parasitizing *S. frugiperda* on a lab scale has even been reported [33,34]. Surveys on the infestation level of *S. frugiperda*, the association between *S. frugiperda* with local parasitoids, and its associated parasitism rate have also been carried out but are limited to a specific period [2,5,35,36,37,38,39,40,41,42]. Preliminary research in Yogyakarta, Indonesia, showed a low attack rate from local parasitoids toward *S. frugiperda* [43]. Thus, there is scattered information regarding the possibility of an association between *S. frugiperda* and local parasitoids. Therefore, this research aimed to study the diversity of parasitoids associated with *S. frugiperda* for three years after it was first discovered in Indonesia. This is important to study as an effort to prepare local biological control agents for potential use in controlling *S. frugiperda*.

## 2. Materials and Methods

### 2.1. Sampling Location Determination

Sampling locations were determined using a purposive sampling method in four central regencies in Yogyakarta, including Sleman, Bantul, Kulon Progo, and Gunung Kidul. Parasitoid sampling activities were carried out in 17 districts of Sleman, 17 districts of Bantul, 12 districts of Kulon Progo, and 18 districts of Gunung Kidul as replication. A total of 2–3 villages were selected from each district as sampling points. From each village, a maize field (±250 m^2^) was chosen as a sampling point using GPS Essentials, resulting in 133 sampling points (Figure 1). Sampling of parasitoids was carried out on maize fields during the vegetative phase (2–3 weeks old).

### 2.2. Sampling

The survey was carried out from January 2020 to May 2022. Sampling was carried out once on each field. Fifty plants per field were used as sample plants with reference to the method by Nonci, Kalqutny, Muis, Azrai, and Aqil [3]. Parasitoids were collected by collecting eggs and larvae of *S. frugiperda*, found on maize plants in every field. Sampling was carried out purposively by taking eggs and larvae found. The samples obtained were brought from the field to the laboratory using an insect-rearing plastic container (21 × 21 cm). The parasitoids from the eggs were placed in an Eppendorf containing 90% ethanol. Meanwhile, the hatched larvae were transferred to and kept individually in plastic cups (400 mL) containing baby corn as a food source until moths or larval/pupal parasitoids emerged. Rearing was maintained under laboratory conditions (26 ± 1 °C, 60–80% r.h.). Parasitoids that emerged were counted, recorded, and grouped based on similar morphological characteristics, then preserved in a 1.5 mL microtube filled with 70% ethanol for further identification.

### 2.3. Parasitoid Identification

The emerging parasitoids were identified at the Plant Protection Laboratory, Department of Agrotechnology, Universitas Muhammadiyah Yogyakarta. Identification of parasitoids was carried out by observing and matching the morphological characteristics of the parasitoids with the relevant literature [39,44,45,46]. The identified parasitoids were photographed at certain magnifications using a Leica S6E Stereo Microscope (Leica Microsystems, Wetzlar, Germany) at the Biological Control Laboratory, Department of Plant Protection, Faculty of Agriculture, IPB University. The parasitoid photos were then processed using TrueChromeII, TCapture 5.1 software (Fuzhou Tucsen Photonics Co., Ltd., Fujian, China) by adjusting the magnification size to the desired unit (mm) to obtain the parasitoid body size. All identified parasitoids were confirmed at the Ecology and Systematics Laboratory, Faculty of Applied Science and Technology, Ahmad Dahlan University.

### 2.4. Data Analysis

A general linear model (GLM) was used to analyze the total number of *S. frugiperda* and the total parasitism rate in each regency. Both data were subjected to stepwise simplification before GLM analysis to determine the appropriate model based on the AIC number. A GLM with Gaussian family and log link function was used to analyze the total number of *S. frugiperda*, while the total parasitism rate was analyzed using a GLM with Gamma family and identity link function. The mean difference of those data between each district was further tested using Tukey’s HSD at a 95% significance level. Areas with zero abundance were not included in the analysis. The GLM analysis was performed using R Statistic version 4.2.2 [47].

The parasitism rate of the egg was observed under the Nikon SMZ18 Stereo Microscope (Nikon Instruments Inc, Melville, NY, USA). Eggs were photographed using Optilab Advance, and the number of eggs was calculated using Image Raster software version 3 (PT. Miconos, Yogyakarta, Indonesia). The parasitism rate was calculated by dividing the number of parasitized hosts by the total number of hosts. Furthermore, parasitoid distribution was mapped based on sampling points (regional administrative data) using ArcGIS 10 (ESRI, Environmental Systems Research Institute, Redlands, CA, USA).

## 3. Results

According to this survey, *S. frugiperda* was discovered to be present in four of Yogyakarta’s central regencies, with significantly different egg population numbers (GLM: F_3,36_ = 7.5369, *p* < 0.001). There was no difference in larva population number across all districts (GLM: F_3,12_ = 0.6335, *p* = 0.6075). The highest egg population of *S. frugiperda* was found in Bantul. The egg population in Gunung Kidul was almost half of Bantul’s population. Meanwhile, the lowest egg populations were found in Kulonprogo and Sleman. Additionally, the parasitism rate of S. frugiperda egg varied significantly amongst districts (GLM: F_3,36_ = 5.2141, *p* < 0.01). The highest parasitism rate of *S. frugiperda* egg in Kulonprogo and Sleman was relatively the same and the lowest in Gunung Kidul, while no significant difference was found in the parasitism rate of larvae (GLM: F_3,12_ = 0.224, *p* = 0.8779, Table 1).

Fifteen species of parasitoids were associated with *S. frugiperda*, and 11 are new reports of parasitoids associated with *S. frugiperda* in Indonesia. Four species were egg parasitoids, eight were larval parasitoids, and three were larval–pupal parasitoids. The egg parasitoids found were Platygasteridae.sp01 (Figure 2a), Platygasteridae.sp02 (Figure 2b), *Trichogramma* sp. (Figure 2c), and *Telenomus remus* (Figure 2d). The larval parasitoids found were Diptera, such as *Archytas marmoratus* (Figure 2e) and *Megaselia* sp. (Figure 2f), and Hymenoptera, such as *Cotesia* sp. (Figure 2g), *Campoletis* sp. (Figure 2h), *Coccygidium* sp. (Figure 2i), Eupelmidae.sp01 (Figure 2j), *Microplitis* sp. (Figure 2k), and *Stenobracon* sp. (Figure 2l). Meanwhile, the larval–pupal parasitoids found were *Brachymeria femorata* (Figure 2m), *B. lasus* (Figure 2n), and *Charops* sp. However, not all parasitoid species were found in every location. Eight species of parasitoids were found in Bantul, seven in Sleman, six in Gunung Kidul, and only three in Kulonprogo.

*Telenomus remus* had the highest parasitism rate (14.74–71.97%). Relatively high parasitism rates were discovered in *Microplitis* sp. and Platygasteridae.sp02. Other parasitoids, such as *Cotesia* sp. and *Stenobracon* sp., had maximum parasitism rates of 16.67% and 11.54%, respectively. *Trichogramma* sp., *Coccygidium* sp., Eupelmidae.sp01, *B. lasus*, and *B. formata* had a maximum parasitism rate of less than 10%; meanwhile, the other parasitoids, including Platygasteridaesp.01, *Campoletis* sp., Megaselia scalaris, *Archytas marmoratus*, and *Charops* sp., had a parasitism rate of less than 5% (Table 2).

A new association was found between *S. frugiperda* and twelve parasitoid species, consisting of three egg parasitoids (Platygasteridaesp.01, Platygasteridaesp.02, and *Telenomus remus*), six larval parasitoids (*Apanteles* sp., *Microplitis* sp., *Campoletis* sp., *Coccygidium* sp., *Eupelmus* sp., and *Stenobracon* sp.), and three larval–pupal parasitoids (*Brachymeria lasus*, *B. femorata*, and *Charops* sp.). *Telenomus remus* has been reported in several regions, including the Western Hemisphere, by an introduction. However, this study reports the first findings of an association between *T. remus* and *S. frugiperda* in Indonesia. Thus, *T. remus* is categorized as a new association, as well as the association between *S. frugiperda* with *Cotesia* sp., *Campoletis* sp., *Microplitis* sp., *Coccygidium* sp., *Stenobracon* sp., *B. lasus*, *B. formata*, and *Charops* sp., because these parasitoids exist elsewhere but not in the Western Hemisphere. Meanwhile, old association was found between *S. frugiperda* and an egg parasitoid (*Trichogramma* sp.) and two larval parasitoids (*Archytas marmoratus* and *Megaselia scalaris*), because these parasitoids were recorded in the original habitat of *S. frugiperda*, the Western Hemisphere.

In contrast to the parasitism rate, the richest and most abundant species of parasitoids were found in Bantul. This amount is far higher than in other places. In total, 8753 parasitoids were obtained in Bantul, 2478 parasitoids in Sleman, 924 in Gunung Kidul, and 479 in Kulon Progo (Table 3).

Based on the population of *S. frugiperda* data and mapping analysis of the parasitoid species distribution found in the field, it can be concluded that S. frugiperda has spread almost throughout the Yogyakarta region (Figure 3). *Telenomus remus* was the most dominant parasitoid because of its abundance. However, *T. remus* was only distributed in a few areas (Figure 4), with the highest abundance found in Bantul.

## 4. Discussion

Our research shows the different numbers of *S. frugiperda* found across all regencies. Bantul has the highest abundance of *S. frugiperda* compared to other locations because of the large cornfield in this area [48]. Corn is also planted in the Kulonprogo and Gunung Kidul, but the corn in these areas is a fodder crop, while corn in Bantul is sweetcorn. *Spodoptera frugiperda* prefers sweet corn to fodder corn [49]. Meanwhile, sweet corn plants in Sleman are sprayed with pesticides more often, resulting in lower *S. frugiperda* populations in this region.

**Figure 4 insects-15-00205-f004:**
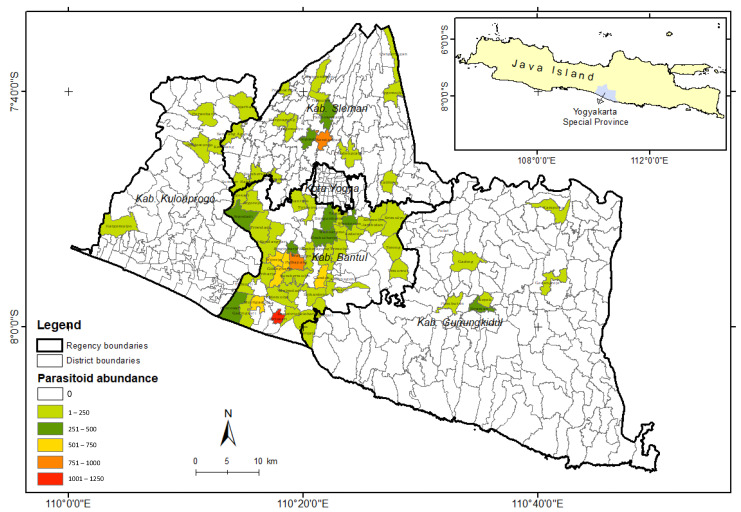
Distribution of *Telenomus remus* in Special Region Yogyakarta, Indonesia. The plus sign is an auxiliary point to indicate the coordinate location.

The abundance and parasitism rates show the opposite, where the highest parasitism rate occurs in Kulonprogo, while the highest abundance is found in Bantul. This happens because of variations in the numbers of pests and parasitized pests overall. Bantul has the highest abundance of parasitoids because of the characteristics of the sampling site, where Bantul Regency serves as Yogyakarta’s primary maize-producing hub [48], making hosts (*S. frugiperda*) more accessible than other districts. According to Kishinevsky et al. [50], an individual parasitoid would be more prevalent in a site if its host population is more numerous, as are its hosts.The results of this research indicate that selecting biological control agents for invasive pests does not always have to be approached by classical biological control methods, which emphasizes that invasive pests are controlled by the natural enemy from the country of origin because local natural enemies cannot control the invasive pest [15], and a new association between invasive pests and local natural enemies will not result in suppression/regulation of the pest because adaptation might take too long. In fact, adaptation can happen relatively quickly, as we found in this study. Elton [51] said that when a parasite species is introduced into an ecosystem with a host or hosts it has never been associated with, the parasite population often rises quickly, and its host population is suppressed.

This study reports 15 parasitoids associated with *S. frugiperda* three years after their discovery in Indonesia. Periodically, only one parasitoid was found associated with *S. frugiperda* in 2019. The number of parasitoid species associated with *S. frugiperda* increased to eight, thirteen, and fifteen species in 2020, 2021, and 2022, respectively. These findings indicate a similar pattern in other areas of Indonesia. For example, one parasitoid was associated with *S. frugiperda* at the beginning of survey activities. According to Jindal et al. [52], no parasitoids were seen in India in 2019 because *S. frugiperda* had recently infected the crop in the late season. Maharani et al. [5] also reported no parasitoids directly associated with *S. frugiperda* in Bandung and Garut, West Java, Indonesia, 2019. However, *A. marmoratus* and *Hymenoptera* larvae were found from *Mythymna separata* obtained from the same field where *S. frugiperda* was collected. Furthermore, Pu’u and Mutiara [53] reported no parasitoids associated with *S. frugiperda* in Ende, East Nusa Tenggara, in 2020. Suroto et al. [40] reported one parasitoid (*Apanteles* sp.) associated with *S. frugiperda* in Banyumas, Central Java, in 2021. Then, Minarni et al. [35] reported the association of *S. frugiperda* with one egg parasitoid (*T. remus*) and three larval parasitoids from the Braconidae, Ichneumonidae, and Chalcididae families in the same location in 2022. Other studies, such as Tawakkal in 2020 [42], reported the association of *S. frugiperda* with six parasitoids in Bogor, West Java. Numerous factors, including the degree of pest infestation [54], geography such as landscape structure [55], and regional variations in agricultural production practices [56], all impact the diversity of parasitoids.

Some parasitoids are similar to those found in the Western Hemisphere, while others differ. *Chelonus insularis* (Hymenoptera: Ichneumonidae) is identified as the primary parasitoid of *S. frugiperda* in most investigations conducted in North, Central, and South America. However, *Eiphosoma laphygmae* (Hymenoptera: Icheumonidae) is more recommended as a prospective candidate for introduction because of its specificity and significance as a parasitoid of the pest across most of its natural habitat [13]. These two parasitoids were not discovered during our research. They were not associated with *S. frugiperda* in other investigations, including those conducted in the Cameroon [57] and India [58].

The most prevalent parasitoid identified in this investigation is *T. remus.* This study reported similar results from other investigations [59,60], where *T. remus* was the dominant parasitoid for *S. frugiperda* because this parasitoid has a high abundance and parasitism rate (14.74–71.97%). Kumela et al. [8] also reported that *T. remus* was a parasitoid of *S. frugiperda* eggs, with the highest parasitism rate (69.3%) in Kenya and Southern China (30–50%) [60]. Sari et al. [33] reported the potential of *T. remus* as a biological agent of *S. frugiperda* in Indonesia, with a parasitism rate of 69.40%. This value is comparable to other egg parasitoids such as *T. chilotraeae* [61]. The parasitism rate of *T. remus* may be higher in a situation with many potential hosts. Junaedi et al. [62] said that the availability of hosts for parasitoid survival could increase the parasitism rate. During sampling in the field, the population of *S. frugiperda* eggs was abundant. The high parasitism rate is also due to *T. remus’* ability to find and recognize its host [63]. Goulart et al. [64] said that the ability of *T. remus* to search and recognize its hosts is better than other *S. frugiperda* egg parasitoids such as *Trichogramma pretiosum*.

*Telenomus remus* is a native egg parasitoid from Malaysia and Papua New Guinea [65]. *Telenomus remus* is a common egg parasitoid used to control pests in the Noctuidae group, especially the *Spodoptera* genus [66]. Studies in Indonesia and other countries like Africa (Benin, Cameroon, Côte d’Ivoire, Ghana, Kenya, Niger, Nigeria, Uganda, South Africa, Tanzania, and Zambia) and Asia (China, India, and Nepal) [67] showed that *T. remus* is a potential biological control agent for controlling *S. frugiperda*. In contrast, Cave [66] showed that introducing *T. remus* has never proved it to be an effective natural enemy in the Americas. These discrepancies between different results render the importance of further investigation.

Other egg parasitoids found were Platygasteridaesp.01 and Platygasteridaesp.02. Morphologically, these parasitoid species differ from *T. remus*, even though they both come from the Platygasteridae family. These parasitoids are characterized by eight flagellum segments and a wider second metasoma segment, different from *T. remus*, which has dilation in the third metasoma [68]. Unfortunately, due to minimal sample conditions, identification could not be carried out to the genus level. Platygasteridae was found, with a lower abundance than *T. remus*. This might happen because the two are not the primary parasitoids of *S. frugiperda* eggs. Platygasteridaesp.01 has a similar morphological character to *Platygaster oryzae*, the main parasitoid of Asian rice gall midge *Orseolia oryzae* [69]. However, the species Platygasteridaesp. 01 could not be confirmed, even though this parasitoid emerged from the rearing of *S. frugiperda* egg clusters collected from maize fields adjacent to rice fields. Meanwhile, Platygasteridaesp.02 has a different body color to *P. oryzae*. *Platygaster oryzae* has a metallic black body color [70,71], while the parasitoids found were a bright yellow. Platygasteridaesp.02 has a fairly high parasitism rate (42%) compared to Platygasteridaesp.01. However, the parasitism incident of Platygasteridaesp.02 was only found in one egg cluster. These two egg parasitoids may not be potential candidates for *S. frugiperda*. Nevertheless, the discovery of two different egg parasitoid species from *T. Remus* and *Trichogramma* indicates the existence of two new associations between *S. frugiperda* and local egg parasitoids.

Another association has been found between *S. frugiperda* and larval parasitoids from Hymenoptera order, including *Microplitis* sp., *Cotesia* sp., *Campoletis* sp., *Coccygidium* sp., Eupelmidaesp.01, and *Stenobracon* sp. *Microplitis* sp., *Cotesia* sp., and *Campoletis* sp. are present in the Western Hemisphere. However, *Cotesia* like *C. ruficrus* are imported from Australia [72], *M. manilae* from Thailand [73], and *C. chloridae* from India [74] to the US. *Microplitis* sp. is a larval parasitoid with the highest parasitism rate (39.7–61.29%) compared to other *S. frugiperda* larval parasitoids. The genus *Microplitis*, reported as a larval parasitoid of *S. frugiperda*, includes *M. manilae* [73]. *Microplitis* is a genera widely distributed throughout all biogeographic zones, with macrolepidopterans as their primary hosts [75]. Moreover, *Cotesia* also has a significant parasitism rate (5.02–16.67%). Supeno et al. [39] and Suroto et al. [40] also reported the incidence of *Cotesia* parasitism on *S. frugiperda* larvae with a 17–22% parasitism rate. Association of *S. frugiperda* larvae with *Microplitis* and *Cotesia* was also reported from Bogor, West Java, with 12.3% and 0.39% parasitism rates, respectively [42]. *Campoletis* had a 0.2–12.69% parasitism rate. *Campoletis* has also been reported in India, with a 2–4% parasitism rate [58]. The results of this study also indicate that *Microplitis*, *Cotesia*, and *Campoletis* are parasitoids of *S. frugiperda* larvae that potentially develop as biological control agents. Surprisingly, *Coccygidium* sp., Eupelmidaesp.01, and *Stenobracon* sp. have never been found in the Western Hemisphere [76]. *Coccygidium* had a 0.2–17.54% parasitism rate. *Coccygidium* was also reported in India, with a 0.001% parasitism rate [58], and in Ghana, with a 3.9–19.3% parasitism rate [44]. *Stenobracon* sp. was also reported from India as the predominant larval parasitoid of *S. frugiperda* [52]. No incidence of parasitism has ever been reported for Eupelmid families in *S. frugiperda.* Thus, a new association was discovered between *S. frugiperda* and larval parasitoid Eupelmidaesp.01.

Apart from the Hymenoptera order, there were also *S. frugiperda* larvae parasitoids from the Diptera order, such as *Archytas marmoratus* and *Megaselia scalaris*. *Archytas marmoratus* is a potential parasitoid for *S. frugiperda* in the American field [77]. However, the parasitism rate found in this study was very low. The parasitism level of *M. scalaris* was also very low. *Megalia scalaris* was first reported in Asia, including India [78] and China [79]. *Megalia scalaris* was also found in the Mexican region [80]. This finding is also the first report on the association between *S. frugiperda* and *M. scalaris* in Indonesia. *Megaselia scalaris* is an insect found in various regions, usually in decaying organic matter [81]. In addition to being reported as a larval parasitoid of *S. frugiperda*, *M scalaris* was also reported to be associated with peach fruit fly, *Bactrocera zonata* (Saunders), Mediterranean fruit fly, *Ceratitis capitata* (Wiedemann) in Egypt [82], and fruit-piercing moths *Thyas coronota* (Fabricius) (Lepidoptera: Erebidea) in India [83]. However, according to a recent study, *M. scalaris* is not recommended as a potential biological control agent for *S. frugiperda. Megaselia scalaris* prefers to consume deceased larvae instead of acting as an endoparasitoid with parasitism rates of 2.2 and 0.7% in third- and fifth-instar larvae of *S. frugiperda*, respectively [84].

In addition to the association between *S. frugiperda* and several egg and larval parasitoids, there was another association with three larval–pupal parasitoids, such as *Brachymeria lasus* and *B. femorata* from the Chalcididae family, and *Charops* sp. from Ichneumonid family. Several chalcidid families reported to be associated with *S. frugiperda* include *B. flavipes* (=robusta) (Fabricius), *B. ovata* (Say), *Conura femorata* (Fabricius), *C. hirtifemora* (Ashmead), *C. igneoides* (Kirby), *C. immaculata* (Cresson), and *C. meteori* (Burks) [76]. These two *S. frugiperda* larval–pupal parasitoids were only found in Bantul. *Brachymeria lasus* and *B. femorata* usually attack hidden insects, such as the banana leafroller caterpillar *Erinota thrax* (Lepidoptera: Hesperiidae) [85], fire caterpillars, and other Noctuidae families. The proximity of banana plants to the sampling site in Bantul Regency suggests that there may be other hosts for these parasitoids. However, these two parasitoids have also been found parasitizing the pupae of *S. frugiperda* in Egypt [86]. Lastly, *Charops* sp. was identified through its pupal characteristics. *Charops* sp. was also reported as a larval–pupal parasitoid of *S. frugiperda* in Cameroon [25], Ghana, and Benin [57].

Our findings support Hokkanen and Pimentel’s theory on using the new association approach for selecting a biological control agent of *S. frugiperda* with local parasitoids such as *T. remus* and *Microplitis* sp. *Telenomus remus* has been used as a biological agent for *S. exigua* [87], and *Microplitis manilae* has also been around for a long time and is associated with other *Spodoptera* species, such as *S. litura,* in Indonesia [88]. This evidence led us to conclude that these parasitoids existed earlier than *S. frugiperda* in Indonesia. Hokkanen and Pimental [18] also said that the original host of the most effective new association biocontrol agents is closely connected to the new host of the agent introduced. Moreover, Kenis [13] also stated that several of the primary parasitoids of *S. frugiperda* within the invaded range are members of the same genera that could be prospective introduction candidates. The original and subsequent hosts have typically belonged to the same genus. Longer taxonomic “jumps” to a new host from distinct families of hosts are also feasible [18], as exemplified by a pupal parasitoid *B. lasus*. Therefore, *T. remus* and *Microplitis* sp. are, thus, possible biological control agent options for use in control initiatives within the new association of biological control concepts equipped to regulate *S. frugiperda*.

Even though this research only reports on the potential of local parasitoids, the results of this research strengthen another approach to selecting biological control agents through a new association. Several studies have shown the success of biological control using local species, such as using several *Trichogramma* species to suppress *S. frugiperda* in China [89]. There are also reports on additional examples, such as the association between the pomegranate whitefly *Siphoninus phillyreae* (Hemiptera: Aleyrodidae) and local parasitoid *Encarsia inaron* (Hymenoptera: Aphelenidae) in Egypt [90], which was successfully controlled with a maximum parasitism rate of 93% with regular release and 36% without release [91]. Contrary to Kenis’s proposed [13], our study argues that a new association between *T. remus* and *S. frugiperda* is a strong candidate for biological control agents in invaded areas. Hence, classical biological control should not be used in a country with abundant *T. remus*, and available data already suggest the ability of *T. remus* to attack and suppress the population of *S. frugiperda*. We strongly advise against introducing *C. insularis* in tropical areas since they are oligophagous, and nontarget effects might disrupt the local food web.

## 5. Conclusions

Fifteen parasitoids are associated with *S. frugiperda*, including twelve new associations. Our study suggests the possibilities of biological control with a new association approach using local parasitoids such as *T. remus* as a reliable strategy to break down the life cycle of *S. frugiperda* before the larvae attack maize fields. Moving forward, further investigations into the scalability in wider agricultural settings will be pivotal for developing sustainable and locally adapted pest control measures.

## Figures and Tables

**Figure 1 insects-15-00205-f001:**
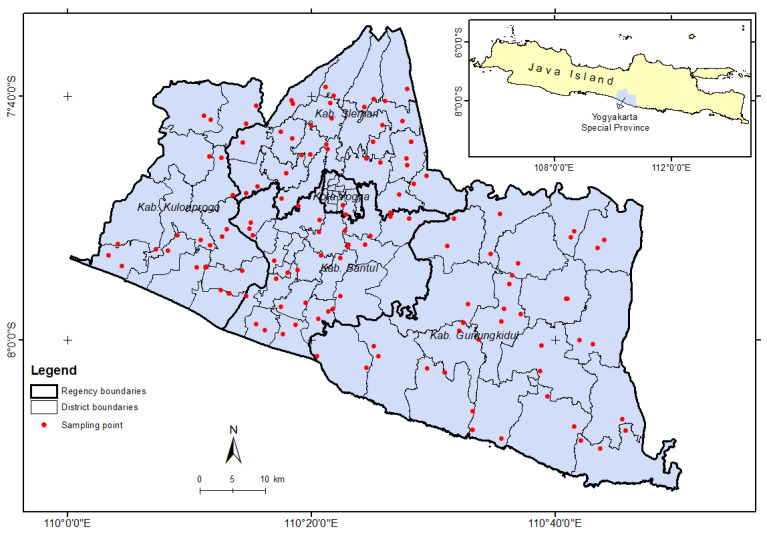
Sampling point of *S. frugiperda* parasitoid in Yogyakarta. The plus sign is an auxiliary point to indicate the coordinate location.

**Figure 2 insects-15-00205-f002:**
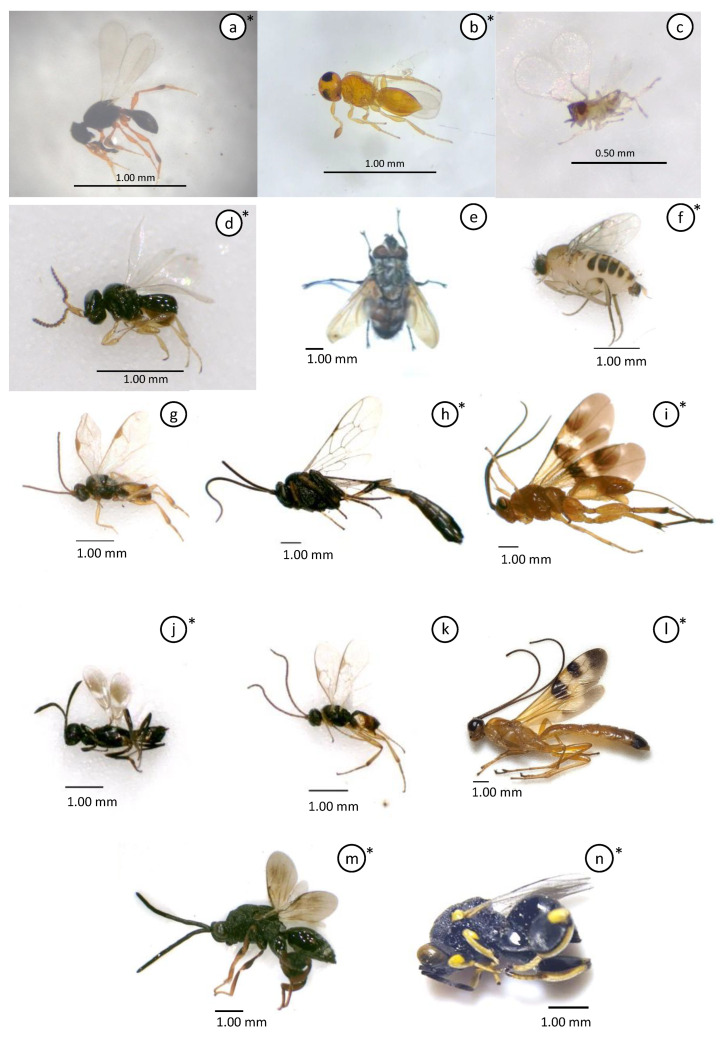
Parasitoid of *Spodoptera frugiperda* in Special Region Yogyakarta, Indonesia. (**a**) Platygasteridae.sp01, (**b**) Platygasteridae.sp02, (**c**) *Trichogramma* sp., (**d**) *Telenomus remus,* (**e**) *Archytas marmoratus*, (**f**) *Megaselia scalaris*, (**g**) *Cotesia* sp., (**h**) *Campoletis* sp., (**i**) *Coccygidium* sp., (**j**) Eupelmidae.sp01, (**k**) *Microplitis* sp., (**l**) *Stenobracon* sp., (**m**) *Brachymeria femorata*, and (**n**) *Brachymeria lasus*. *: New report.

**Figure 3 insects-15-00205-f003:**
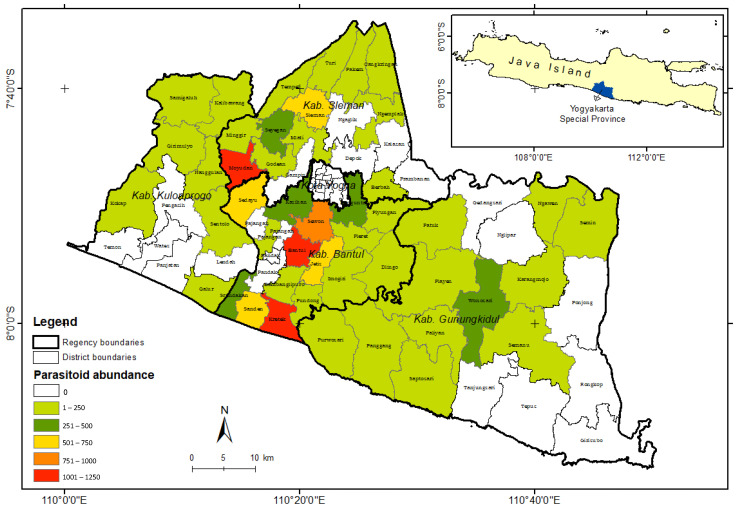
Distribution of *Spodoptera frugiperda*’s parasitoid in Special Region Yogyakarta, Indonesia. The plus sign is an auxiliary point to indicate the coordinate location.

**Table 1 insects-15-00205-t001:** Number of *S. frugiperda* (egg and larvae) and total parasitism rate in different districts of Yogyakarta, Indonesia.

District	Number of *S. frugiperda* (Mean ± SE)	Parasitism Rate (%) (Mean ± SE)	n
Egg			
Bantul	1918.00 ± 465.18 b	28.10 ± 6.81 b	17
Gunung Kidul	997.14 ± 376.88 ab	10.38 ± 3.92 a	7
Kulonprogo	146.00 ± 65.29 a	71.97 ± 32.18 ab	5
Sleman	190.09 ± 57.31 a	34.03 ± 10.26 ab	11
Larva			
Bantul	43.20 ± 21.60 a	5.34 ± 2.67 a	4
Gunung Kidul	40.50 ± 20.25 a	18.39 ± 9.20 a	4
Kulonprogo	14.00 ± 9.90 a	11.45 ± 8.10 a	2
Sleman	48.50 ± 19.80 a	10.55 ± 4.31 a	6

Means with different letters in a column are significantly different by Tukey HSD test (α = 0.05). SE: standard error. n: replication.

**Table 2 insects-15-00205-t002:** Parasitism rate of *Spodoptera frugiperda*’s parasitoid in Special Region Yogyakarta, Indonesia.

Order	Family	Species	Parasitism Rate (Mean ± SE, %)			
			Bantul	Gunung Kidul	Kulon Progo	Sleman
Egg parasitoid						
Hymenoptera	Platygasteridae	Platygasteridae.sp01 *		1.46 ± 1.01		
	Platygasteridae	Platygasteridae.sp02 *				42.00 ± 0.00
	Platygasteridae	*Telenomus remus* *	37.29 ± 4.16	14.74 ± 5.58	71.97 ± 15.62	33.23 ± 8.33
	Trichogrammatidae	*Trichogramma* sp.	5.78 ± 4.36			
Larval parasitoid						
Hymenoptera	Braconidae	*Cotesia* sp. *		11.89 ± 7.30	16.67 ± 0.00	5.02 ± 0.79
	Ichneumonidae	*Campoletis* sp. *		1.61 ± 0.00		2.78 ± 0.00
	Braconidae	*Coccygidium* sp. *		5.56 ± 0.00		2.78 ± 0.00
	Eupelmidae	Eupelmidae.sp01 *			6.25 ± 0.00	
	Braconidae	*Microplitis* sp. *		39.70 ± 26.92		61.29 ± 0.00
	Braconidae	*Stenobracon* sp.*	11.54 ± 0.00			
Diptera	Tachinidae	*Archytas marmoratus*	4.03 ± 0.22			
	Phoridae	*Megaselia scalaris*	3.70 ± 0.00			
Larval–pupal parasitoid						
Hymenoptera	Chalcididae	*Brachymeria femorata* *	6.25 ± 0.00			
	Chalcididae	*Brachymeria lasus* *	6.82 ± 0.00			
	Ichneumonidae	*Charops* sp. *	1.04 ± 0.00			4.02 ± 0.87

*: New association; SE: standard error.

**Table 3 insects-15-00205-t003:** Diversity of *Spodoptera frugiperda*’s parasitoid in Special Region Yogyakarta, Indonesia.

Order	Family	Species	Abundance			
			Bantul	Gunung Kidul	Kulon Progo	Sleman
Egg parasitoid						
Hymenoptera	Platygasteridae	Platygasteridae.sp01		49		
	Platygasteridae	Platygasteridae.sp02				21
	Platygasteridae	*Telenomus remus*	8536	831	466	2324
	Trichogrammatidae	*Trichogramma* sp.	1988			
Larval parasitoid						
Hymenoptera	Braconidae	*Cotesia* sp.		27	12	110
	Ichneumonidae	*Campoletis* sp.		1		1
	Braconidae	*Coccygidium* sp.		1		1
	Eupelmidae	Eupelmidae.sp01			1	
	Braconidae	*Microplitis* sp.		15		19
	Braconidae	*Stenobracon* sp.	3			
Diptera	Tachinidae	*Archytas marmoratus*	2			
	Phoridae	*Megaselia scalaris*	5			
Larval–pupal parasitoid						
Hymenoptera	Chalcididae	*Brachymeria femorata*	5			
	Chalcididae	*Brachymeria lasusi*	3			
	Ichneumonidae	*Charops* sp.	1			2
Species richness			8	6	3	7
Total abundance			8753	924	479	2478

## Data Availability

The data supporting this study’s findings are available on request from the author (I.N.).

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
