# Peer review of "Association of a Global Invasive Pest Spodoptera frugiperda (Lepidoptera: Noctuidae) with Local Parasitoids: Prospects for a New Approach in Selecting Biological Control Agents"

_insects, 2024, doi:10.3390/insects15030205_

Round 1

Reviewer 1 Report (Previous Reviewer 1)

Comments and Suggestions for Authors

The authors have given satisfactory answers to all my comments. I have no further queries.

However, I would just recomment the authors in their conclusion (no "s"/plurial is needed) to formulate different strategies about biocontrol. In particular, what parasitoids they recommend to use on S.frugiperda. In my experience it would be interesting to suggest field releases of egg parasitoids (Trichogramma + Telenomus) as a reliable strategy to break down the cycle of this pest before the larvae attack maize fields. 

Author Response

Response to Reviewer’s Comments

REVIEWER 1

The authors have given satisfactory answers to all my comments. I have no further queries.

However, I would just recomment the authors in their conclusion (no "s"/plurial is needed) to formulate different strategies about biocontrol. In particular, what parasitoids they recommend to use on S.frugiperda. In my experience it would be interesting to suggest field releases of egg parasitoids (Trichogramma + Telenomus) as a reliable strategy to break down the cycle of this pest before the larvae attack maize fields.

Response. Thank you for your help reviewing our manuscript and your appreciation of our revision. We revised the conclusion. However, we can only recommend Telenomus based on the available data.

Reviewer 2 Report (Previous Reviewer 2)

Comments and Suggestions for Authors

Author Response

Response to Reviewer’s Comments

REVIEWER 2

This work has many weaknesses, some of them already highlighted on the previous review. The modifications don't respond to those criticisms. On the contrary some modifications go on the wrong direction. The title is still ambiguous, and the objectives aren't clear. There are big failures on the methodology and on the results. Even if I agree that is very important issue concerning the biological control of this pest, I think this paper has not the quality necessary to be published on Insects. I advise the authors to conceive a simpler paper if the data available is not sufficient to reformulate this paper. I highlight in yellow the new comments to this paper.

Response: Thank you for your critical opinion regarding our manuscript. The modification was made based on the other reviewer’s comment. They ask to show the population number to compare the parasitism rate. We revised the title and objective after these modifications and revisions. We revised the data in Table 1.

Title

Prospects for a new approach in biological control

If the article talks about a "new approach" in the introduction, it should be explained what the

"old approach" is with examples and what the "new approach" is, related to this pest.

Otherwise, the title doesn't make much sense. If the new approach is augmentative biological control, it is not new, and it is already implemented to this pest with T. Remus.

Response. Thank you for your input. During revision, we realized that the new approach is used for selecting biocontrol agents. Thus, we revised the title as presented in this revised version.

Introduction

Much of the introduction is devoted to classical biological control when that is not the purpose of the article. Why there is not an attempt to document more examples of biological control of introduced pests with local fauna in other regions? (not changed or explained)

Response. We devote classical biological control as the common approach to selecting biological control. While the new association is a new approach that is rarely studied. We thank you for your critical opinion, and now we have revised the introduction based on your input as described in the second and third paragraphs. I hope this explanation now will be easy to understand by the readers.

Line 87: remove "their"; Before the introduction of the pest they are not enemies of the pest.

Response: Revised in line 73

  1. Materials and Methods

Line 109-111 - this sentence is not part of this section.

Response: Removed

Lines 142-143 - "existing associations when a parasitoid was not originally from but currently present in Western Hemisphere. "That last sentence doesn't make much sense. Existing associations are all those that have already been documented.

Response. We revised the association category as described in line 144-150

  1. Results

Tabela 1 - The table does not clarify the results, but rather raises new questions. In order to be able to make comparisons between the various zones, it is necessary to explain the monitoring protocol in detail. The same number of plants must be observed in each region/sampling point and eggs, larvae and pupae must not be mixed. The eggs are laid in groups, each containing about 100 - 200 eggs, while the larvae and pupae are dispersed. It doesn't make much sense in terms of comparisons to give the same weight to an egg or a larva. When we find, we usually find 100 - 200 eggs and eventually several parasitized eggs. When we find a larva, we may eventually find 1 parasitoid (or several if there is polyembryony). The sampling protocol must be adequately explained in the material and methods and the treatment of data, results and must reflect this protocol. Also, the discussion must of the results must reflect the protocol.

Response. Thank you for the input. The sampling protocol was described in the method session. We don’t have the data on the number of plant samples. However, we added the average area of a sampling site in line 114 and separated the data based on the host stage, as shown in Table 1.

Table 2 - If there is an average, the deviation should be indicated.

Response: Revised by adding the standard error

Figure 3 and 4: In the legend and figure it should be clear whether "no data" means that there was no sampling or there was a sampling but the result is zero.

Response: Revised to zero. There was a sampling but the result is zero.

In order to properly understand and discuss the differences observed, it is necessary to complete the information: from each region how many stages (eggs, larvae and pupae) were brought to the laboratory?

What is the percentage of parasitoid emergence from each state in each region. On the other hand, discussing the abundance of the various groups of parasitoids through the material emerged in the laboratory will put at an advantage parasitoids that attack the eggs because when the eggs are collected there are naturally hundreds of eggs that are collected. Also, only one parasitoid develop on each pest or many larvae may develop within the pest.

Response: Revised in Table 1

Table 1 does not answer the questions raised. On the contrary, you go in the opposite direction, mixing everything.

Response: Revised the data based on the host stage as shown in revised table 1

209 - what is "often to outbreak levels" for a parasite?

Response: for a pest

It was not corrected in the text

Response: Revised in line 264

Lines 254-271 - You can not show an increase, comparing with other regions.

Response: Revised in line 268

Line 379 - If T. remus is not a new association, how does this study support the theory on using the New Association biological control approach?

Response: Because this study first reported the association between T. remus and S. frugiperda in Indonesia. We then regenerate the association category as described in line 134-139

Reviewer 3 Report (Previous Reviewer 3)

Comments and Suggestions for Authors

After reviewing the revised version of the manuscript, I am confident that the document has undergone substantial improvements and is now suitable for publication in the journal Insects.

Author Response

Response to Reviewer’s Comments

REVIEWER 3

After reviewing the revised version of the manuscript, I am confident that the document has undergone substantial improvements and is now suitable for publication in the journal Insects.

Response. Thank you for your help reviewing our manuscript and your appreciation of our revision.

Round 2

Reviewer 2 Report (Previous Reviewer 2)

Comments and Suggestions for Authors

My last comments are highlighted in yellow on the file Cover leter 2.

Author Response

Response to Reviewer’s Comments

REVIEWER 2

My last comments are highlighted in yellow.

General Comments

The paper has greatly improved but I think some corrections are still required.

  1. Lines 144-150. New and old associations definitions (clearer); (comments below)
  2. Table 1 – numbers of S. frugiperda (comments below)
  3. Lines 254-271 (comments below)
  4. Line 269 (comments below)
  5. Line 273 (comments below)
  6. Clarifying the status of T. remus (comments below)

I consider accepted pending minor revisions

Response: Thank you for your appreciation, and continue carefully checking our manuscript. We have revised the manuscript based on above mentioned comment with the responses as follow.

Table 1

Author's Notes

Response to Reviewer’s Comments

REVIEWER 2

This work has many weaknesses, some of them already highlighted on the previous

review. The modifications don't respond to those criticisms. On the contrary some

modifications go on the wrong direction. The title is still ambiguous, and the objectives

aren't clear. There are big failures on the methodology and on the results. Even if I agree

that is very important issue concerning the biological control of this pest, I think this paper

has not the quality necessary to be published on Insects. I advise the authors to conceive

a simpler paper if the data available is not sufficient to reformulate this paper. I highlight in

yellow the new comments to this paper.

Response: Thank you for your critical opinion regarding our manuscript. The modification

was made based on the other reviewer’s comment. They ask to show the population

number to compare the parasitism rate. We revised the title and objective after these

modifications and revisions. We revised the data in Table 1.

I understand the importance of the number of S. frugiperda collected. For the parasitism

rate it’s OK to analyse the differences because it is a ratio which is independent on the

number. But you cannot compare the number of insects collected between regions unless

the same sampling protocol is adopted in each sampling site. If you say that all plants

were observed in the (± 250 m2), ok, you can compare the numbers per area. Even if all the

plants were sampled in this area it is still indicated in the text “± “and probably the plant

density will be different at each sampling point. Thus, there are many errors associated

with this comparison. Likewise, for the same reasons it is not correct to indicate an

average and a deviation. If you cannot standardize the sampling, It would be preferable to

provide the number of S. frugiperda collected with the median together with the minimum

and maximum values, without making any kind of comparison.

In any case, the methodology should reflect the choices made.

Response: Thank you for carefully reviewing the data presentation and method. We apologize for not including complete references in the sampling section. We used 50 plant samples per field based on standard methods from the Indonesian Ministry of Agriculture, as described in lines 121-122. Thus, the data can be compared between one location and another.

Lines 144-150

Associations between S. frugiperda and parasitoids are defined as new associations when

a parasitoid has never been reported in the Western Hemisphere and another region,

relatively new associations when a parasitoid was not originally from but is currently

present in the Western Hemisphere, considered new associations when a parasitoid was

present in other regions but not in the Western Hemisphere, and old association when a

parasitoid was recorded in the Western Hemisphere. This grouping uses the Western

Hemisphere as a point of reference for S. frugiperda 's original habitat.

These definitions should be clearer. The definition of “new association” appears twice.

Response: Actually, we have four definitions. New association, relatively new association, considered new association, and old association as described in line146-151.

Tabela 1 - The table does not clarify the results, but rather raises new questions. In order to

be able to make comparisons between the various zones, it is necessary to explain the

monitoring protocol in detail. The same number of plants must be observed in each

region/sampling point and eggs, larvae and pupae must not be mixed. The eggs are laid in

groups, each containing about 100 - 200 eggs, while the larvae and pupae are dispersed. It

doesn't make much sense in terms of comparisons to give the same weight to an egg or a

larva. When we find, we usually find 100 - 200 eggs and eventually several parasitized

eggs. When we find a larva, we may eventually find 1 parasitoid (or several if there is

polyembryony). The sampling protocol must be adequately explained in the material and

methods and the treatment of data, results and must reflect this protocol. Also, the

discussion must of the results must reflect the protocol.

Response. Thank you for the input. The sampling protocol was described in the method

session. We don’t have the data on the number of plant samples. However, we added the

average area of a sampling site in line 114 and separated the data based on the host stage,

as shown in Table 1.

My comments on table 1 are above.

Response. See above.

Line 269 _ often to outbreak levels of a pest, I think this fragment must be removed, it

doesn’t make much sense.

Response. Removed.

Line 273 The result shows that there are more parasitoids associated with S. frugiperda.

Response: More then what?

Lines 254-271 - You can not show an increase, comparing with other regions.

Response: Revised in line 268

Response: Line 268 doesn’t address this question

Response. We mean that we found an increasing number of species yearly over the 3-year study period, as described in lines 269-272.

Line 379 - If T. remus is not a new association, how does this study support the theory on

using the New Association biological control approach?

Response: Because this study first reported the association between T. remus and S.

frugiperda in Indonesia. We then regenerate the association category as described in line

134-139.

Response: But since it is already reported in China, parasitising S.frugiperda it should not

be considered a new association, even in the sense you defined it.

Similarly, T. remus was found in Southern parts of Asia, with FAW egg mass parasitism

rates of 30% already in the first season after pest arrival in China (Liao et al. 2019).

(doi.org/10.1186/s43170-021-00071-6)

From you: old association is when a parasitoid was recorded in the Western Hemisphere.

Response. Although T.remus is reported elsewhere. It’s a new finding in Indonesia. So, we consider this as a new, or relatively new association because it is already existing elsewhere. Furthermore, the association of T. remus with S. frugiperda has never occurred in its native area. The presence of T.remus in the western hemisphere originates from the introduction and is not established. We also revised the meaning of old association to when a parasitoid was a native to the Western Hemisphere.

Round 3

Reviewer 2 Report (Previous Reviewer 2)

Comments and Suggestions for Authors

Currently, my only concern regarding this article is the definitions of associations.

“considered new association” and “relatively new associations “. These definitions are inappropriate because they lend themselves to confusion. They can resemble sentence mistakes. The following sentences are practically the same but indicate very different situations according to your definitions.

T. remus is considered a new association.

T remus is a considered new association.

T remus is a relatively new associations, can give a temporal idea of the association.

In fact, there are only two types of associations. Known and unknown. The known ones can be natural if the host and the parasitoid are naturally in the same region or, on the other hand, are the result of the introduction of the host or the parasitoid. However, it can be indicated new association to Indonesia, new association to India, new association to America..... It is clearer than publishing definitions that lead themselves to confusion.

Apart from this issue, the article should be published in its current state.

Author Response

Response to Reviewer’s Comments

Currently, my only concern regarding this article is the definitions of associations.

“considered new association” and “relatively new associations “. These definitions are inappropriate because they lend themselves to confusion. They can resemble sentence mistakes. The following sentences are practically the same but indicate very different situations according to your definitions.

T remus is considered a new association.

T remus is a relatively new associations, can give a temporal idea of the association.

In fact, there are only two types of associations. Known and unknown. The known ones can be natural if the host and the parasitoid are naturally in the same region or, on the other hand, are the result of the introduction of the host or the parasitoid. However, it can be indicated new association to Indonesia, new association to India, new association to America....It is clearer than publishing definitions that lead themselves to confusion.

Apart from this issue, the article should be published in its current state.

Response: Thank you for the input. We agree with the reviewer’s feedback and have removed the irrelevant grouping of new association definitions. As a result, there are minor changes to the abstract section in lines 33-37 and results in lines 198-208. We hope our manuscript is now improved and acceptable for publication.

This manuscript is a resubmission of an earlier submission. The following is a list of the peer review reports and author responses from that submission.

Round 1

Reviewer 1 Report

Comments and Suggestions for Authors

Thank you for this good article. However I have some comments about the parasitoids found :

1- Apanteles is an old genus name : the current name is *Cotesia* so please double check this name in your manuscript and change it.

2- I am very skeptical about the diptera parasitoid found on Sopodoptera frugiperda (Phoridae) Megaselia scalaris. This insect is not known as a larval parasitoid of Lepidoptera. It is an omnivorous fly feeding on a variety of fermenting and decomposing substances and can be responsible for some human diseases. So it was a big surprise to mention this fly as parasitoid. The only fly that is well known to parasitize Lepidoptera is Tachinid fly (Tachinidae) and you found one species which is OK. M.scalaris could have been found by mistake on dejections (frass) of Spodoptera larvae...but not inside the larvae. in addition you found on very low numbers...so I have doubt that this fly is a real parasitoid that can be proposed in biocontrol programmes.

3- How could you explain such a high number of parasitoids in Bantul compare to other places. I know that you gave an explanation but this is surprising. Do the farmers use insecticides in the other places surveyed? 

4 - It is a pity to my opinion to mention only "unknow Hymenoptera". There are some taxonomists around the world in Europe, in Australia, in America that are able to identify these species. Why not going a little bit deeper to get the ID of these parasitoids. 

Comments on the Quality of English Language

Please double check the plurial of names sometimes missing in your manuscript (sampling point/sampling points, parasitoids/parasitoid, natural enemy, natural enemies...) : this is particularly for the caption of tables and figures 

Reviewer 2 Report

Comments and Suggestions for Authors

Reviewer 3 Report

Comments and Suggestions for Authors

A review report:

The study presented by the researchers focuses on investigating the potential of local parasitoids as biological control agents against the globally invasive pest, Spodoptera frugiperda. The paper addresses an important aspect of classical biological control theory, challenging the notion that effective control of invasive pests necessitates natural enemies from their area of origin.

The authors conducted a rapid survey in Yogyakarta, Indonesia, spanning four districts over a three-year period, aiming to identify parasitoid species associated with S. frugiperda. Their findings reveal a significant increase in parasitoid species richness over the study period, with a total of 15 parasitoid species identified. Notably, the study detected four egg parasitoids, eight larval parasitoids, and three pupal parasitoids associated with S. frugiperda, indicating a diverse array of natural enemies targeting different developmental stages of the pest.

One of the key highlights of the research is the dominance of Telenomus remus among the identified parasitoids, displaying higher abundance and parasitism rates compared to others. Additionally, the discovery of a new association between S. frugiperda and twelve parasitoid species, including previously unreported associations with certain egg, larval, and pupal parasitoids, adds significant depth to the understanding of this pest-parasitoid relationship.

Notably, the study reports the first-ever association of S. frugiperda with the larval parasitoid Megaselia scalaris in Indonesia, shedding light on previously unknown interactions between the pest and local natural enemies.

The implications of these findings challenge the conventional principles of classical biological control, suggesting that local natural enemies can adapt swiftly to invasive pests. This observation potentially opens up avenues for exploring the efficacy of local parasitoids in controlling S. frugiperda, thereby offering promising insights into alternative biological control strategies.

However, the MS could benefit from additional details, such as the ecological dynamics and interactions between the identified parasitoids and S. frugiperda, as well as a more comprehensive discussion on the practical applications of these findings in pest management strategies.

In summary, the study contributes significantly to the field by highlighting the potential of local parasitoids as effective biological control agents against S. frugiperda, challenging existing paradigms in classical biological control theory. The new associations identified in this research pave the way for further exploration and practical utilization of these local natural enemies for managing this highly invasive pest.

Comments to authors:

The Materials and Methods section of the manuscript presents a comprehensive approach to sampling and identifying parasitoids associated with S. frugiperda. However, there are some potential shortcomings that could be addressed:

Sampling Location Determination: While the study outlines the use of a stratified random sampling method across four central districts, providing greater geographical coverage, it lacks details on the specific criteria used for the selection of villages within each sub-district or maize fields within villages. Clarifying the selection criteria could enhance the reproducibility and validity of the sampling strategy.

Sampling Frequency and Replication: The sampling design involved a single sampling event at each field. Considering the temporal variability in parasitoid populations, conducting multiple samplings over time or replicating sampling within each field could provide a more robust understanding of parasitoid diversity and abundance, reducing the potential impact of stochastic events on results.

Sampling Procedure and Identification: The description of the sampling process is comprehensive but lacks specific information on the duration of each sampling event and whether different life stages of S. frugiperda (eggs, larvae, and pupae) were equally sampled or if there was any bias in collection among these stages. Additionally, while morphological identification is described, the absence of molecular or genetic techniques for species confirmation might limit the accuracy of the identified parasitoids.

Data Analysis: The description of data analysis, particularly the use of Microsoft Office Excel for tabulating diversity and parasitism rates, lacks detail on the specific statistical analyses conducted. Providing information on the statistical tests employed for analyzing diversity and parasitism rates would enhance the transparency and rigor of the analysis.

Documentation of Parasitoids: The manuscript describes the documentation and measurement of identified parasitoids using specific software and equipment. However, details on the specific metrics measured and the methodology used for measurements are not provided, which could potentially hinder reproducibility.

Addressing these points by providing additional clarity and detail would strengthen the methodological rigor of the study and facilitate a better understanding of the sampling and identification processes in assessing parasitoid diversity associated with S. frugiperda.

Final comment: The authors have executed commendable work in this research. A minor revision is recommended.